# RTA dh404 Induces Cell Cycle Arrest, Apoptosis, and Autophagy in Glioblastoma Cells

**DOI:** 10.3390/ijms24044006

**Published:** 2023-02-16

**Authors:** Tai-Hsin Tsai, Yu-Feng Su, Cheng-Yu Tsai, Chieh-Hsin Wu, Kuan-Ting Lee, Yi-Chiang Hsu

**Affiliations:** 1Division of Neurosurgery, Department of Surgery, Kaohsiung Medical University Hospital, Kaohsiung 807378, Taiwan; 2Department of Surgery, School of Medicine, College of Medicine, Kaohsiung Medical University, Kaohsiung 807378, Taiwan; 3Graduate Institutes of Medicine, College of Medicine, Kaohsiung Medical University, Kaohsiung 807378, Taiwan; 4Division of Neurosurgery, Department of Surgery, Kaohsiung Municipal Ta-Tung Hospital, Kaohsiung 80145, Taiwan; 5School of Medicine, I-Shou University, Kaohsiung 82445, Taiwan

**Keywords:** apoptosis, autophagy, cell cycle, RTA dh404, glioblastoma

## Abstract

RTA dh404 is a novel synthetic oleanolic acid derivative that has been reported to possess anti-allergic, neuroprotective, antioxidative, and anti-inflammatory properties, and exerts therapeutic effects on various cancers. Although CDDO and its derivatives have anticancer effects, the actual anticancer mechanism has not been fully explored. Therefore, in this study, glioblastoma cell lines were exposed to different concentrations of RTA dh404 (0, 2, 4, and 8 µM). Cell viability was evaluated using the PrestoBlue™ reagent assay. The role of RTA dh404 in cell cycle progression, apoptosis, and autophagy was analyzed using flow cytometry and Western blotting. The expression of cell cycle-, apoptosis-, and autophagy-related genes was detected by next-generation sequencing. RTA dh404 reduces GBM8401 and U87MG glioma cell viability. RTA dh404 treated cells had a significant increase in the percentage of apoptotic cells and caspase-3 activity. In addition, the results of the cell cycle analysis showed that RTA dh404 arrested GBM8401 and U87MG glioma cells at the G2/M phase. Autophagy was observed in RTA dh404-treated cells. Subsequently, we found that RTA dh404-induced cell cycle arrest, apoptosis, and autophagy were related to the regulation of associated genes using next-generation sequencing. Our data indicated that RTA dh404 causes G2/M cell cycle arrest and induces apoptosis and autophagy by regulating the expression of cell cycle-, apoptosis-, and autophagy-related genes in human glioblastoma cells, suggesting that RTA dh404 is a potential drug candidate for the treatment of glioblastoma.

## 1. Introduction

Glioblastoma multiforme (GBM) is a lethal central nervous system malignancy with a median patient survival of 8–15 months [1,2]. Currently, due to complex signaling pathway activation, there is no cure for GBM, and the first-line drug temozolomide (TMZ) can only prolong the life expectancy of treated patients [3]. Therefore, the combined treatment of surgery and radiation therapy seems to be an interesting approach for treating GBM. The Stupp protocol [4], which established the standard of care for newly diagnosed GBM patients younger than 65 years of age with good performance status, remains unsatisfactory. Therefore, there is an urgent need to develop new drugs or biomolecules with different mechanisms of action for combined therapy, which could not only reduce the concentration of a single drug required to obtain the same therapeutic effect but also increase the biological activity of the drug improving therapeutic potential. In addition, the side effects may be significantly reduced.

CDDO (2-cyano-3,12-dioxooleana-1,9(11)-dien-28-oic acid) was chemically modified with oleanolic acid as a framework to obtain oleanolic acid-derived synthetic triterpenoids [5]. Oleanolic acid, a pentacyclic triterpenoid widely found in plants, has weak biological activity [6,7]. Therefore, after a series of chemical modifications, synthetic oleanane triterpenoids have enhanced solubility, bioavailability, and potency with active biological pleiotropic effects [7]. Synthetic oleanane triterpenoids are anti-inflammatory and cytoprotective, induce tumor cell differentiation, and suppress tumor cell growth [8]. Recently, there has been increased interest in the pleiotropic effects of synthetic oleanane triterpenoids in various diseases, particularly cancer. Owing to the pleiotropic effects of their biological activity, synthetic oleanane triterpenoids can be used in the prevention and treatment of cancer [8] (Figure 1).

RTA dh404 is a synthetic oleanane triterpenoid compound that potently activates Nrf2 and inhibits the pro-inflammatory transcription factor, NF-κB. RTA dh404 increased Nrf2 expression, decreased oxidative stress, decreased NF-κB, and decreased the levels of proinflammatory mediators [9,10]. Animal experiments demonstrated that it has therapeutic effects on the kidney, liver, heart, pancreas, and metabolic diseases. First, RTA dh404 has a protective effect against kidney disease, reducing glomerular sclerosis, renal interstitial inflammation, and fibrosis in mice with chronic kidney disease [11]. Experiments in mice with nephrotic syndrome confirmed that RTA dh404 has a renoprotective effect by preventing mitochondrial dysfunction in renal tubular epithelial cells [2]. Furthermore, RTA dh404 has a protective effect against cardiac disease, significantly inhibits oxidative stress in the heart, inhibits pathological cardiac remodeling and dysfunction, and reduces mortality by inhibiting oxidative stress in cardiomyocytes [12,13]. In addition, RTA dh404 has a protective effect on hepatopancreatic diseases. It regulates liver cytoprotective genes and maintains bile flow [14]. It also attenuates acute pancreatitis by reducing oxidative stress and pro-inflammatory mediators [15]. Finally, RTA dh404 is also protective against metabolic diseases such as diabetes and obesity and is well tolerated and effective in rodent models of type 2 diabetes and obesity [16].

RTA dh404 has similar dose-dependent pleiotropic effects as synthetic oleanane triterpenoids, with antioxidant and anti-inflammatory effects at low doses and anticancer effects at high doses. Herein, we report, for the first time, the potential anticancer effect of RTA dh404 on human GBM cell lines. In the present study, we aimed to explore the anticancer effect of RTA dh404, a novel synthetic oleanane triterpenoid derivative, on human glioblastoma cells in vitro and elucidate the underlying mechanisms of its action. RTA dh404 effectively decreased the viability of GBM8401 and U87MG cells. Furthermore, we demonstrated enhanced apoptosis, autophagy induction, and cell cycle arrest in RTA dh404-treated GBM cells.

## 2. Results

### 2.1. RTA dh404 Inhibited the Viability of GBM8401 and U87MG Cells in a Time- and Dose-Dependent Manner

To determine whether RTA dh404 is a potential selective anticancer agent, human brain malignant glioma cells (GBM8401 and U87MG) were treated with different concentrations of RTA dh404 (0, 0.25, 0.5, 0.75, 1, 2, 4, or 8 µM) for 24 h and cell viability was measured using the PrestoBlue™ reagent assay. As shown in (Figure 2A,B), RTA dh404 significantly decreased the viability of both GBM8401 and U87MG human brain malignant glioma cells, which suggests that RTA dh404 is a selective anticancer agent against human brain malignant glioma cells. Furthermore, the viability of GBM8401 and U87MG cells after treatment with RTA dh404 does not only decrease in a “dose-dependent manner”, but also in a “time-dependent manner” indicating that RTA dh404 dose-dependently inhibited clone formation in GBM8401 and U87MG cells. The effect of RTA dh404 on cell growth was assessed by colony formation assay in GBM8401 and U87MG cells, which showed that RTA dh404 significantly inhibited colony formation compared to the control (Figure 2D).

### 2.2. RTA dh404 Induced Caspase-Dependent Apoptosis in GBM8401 and U87MG Cells

To confirm these findings associated with apoptosis, we analyzed cell death using Annexin V-FITC and PI staining to confirm whether RTA dh404 caused cytotoxic effects on glioblastoma cells. This evaluation revealed an increase in GBM8401 cell death after treatment with RTA dh404 (0, 2, 4, or 8 µM) for 24 h (Figure 3). Cell populations and percentages of cells undergoing apoptosis were measured using flow cytometry. The results of the Annexin-FITC/PI assay showed that apoptotic cells with or without RTA dh404 treatment demonstrated significant changes. Compared to untreated cells, the percentage of apoptotic cells was significantly increased in the treated cells (Figure 3B), and the results showed that RTA dh404 can induce glioblastoma apoptosis but not necrosis.

### 2.3. RTA dh404 Increased the Amount of ACTIVE CASPASE-3 in Glioma Cells

Subsequently, a caspase-3 assay was performed to determine whether cell death was triggered by caspase-dependent apoptosis. As the concentration of RTA dh404 increased (0–8 µM), the number of active caspase-3 positive cells increased linearly. A significant increase in caspase-3 activity was detected in RTA dh404-treated cancer cells (Figure 4, y = 2.761x − 1.69, R² = 0.8512). We suggest that RTA dh404 induces GBM8401 cell death via caspase-3-dependent apoptosis. Taken together, our data indicate that the inhibition of GBM8401 and U87MG cell viability by RTA dh404 was due to its capacity to induce caspase-dependent apoptosis.

### 2.4. RTA dh404 Induced the Accumulation of G2/M-Arrested Glioma Cells

To determine whether cell cycle arrest was induced by RTA dh404, GBM8401 and U87MG cells were treated with RTA dh404 and cell cycle progression was examined by PI staining to measure cell cycle distribution. When cells were exposed to different concentrations of RTA dh404 (0, 2, 4, or 8 µM), we observed that RTA dh404 increased the number of cells in the G2/M phase (Figure 5). Our results indicate that exposure to RTA dh404 resulted in a dose-dependent increase in the number of cells in the G2/M phase, suggesting a reduction in mitosis in GBM8401 and U87MG cells. The results indicated that RTA dh404 inhibited GBM8401 and U87MG cell proliferation, which was associated with cell cycle arrest in the G2/M phase.

### 2.5. RTA dh404 Modulated the Expression of Apoptosis-, Autophagy-, and G2/M Cell Cycle-Associated Proteins in Glioma Cells

To investigate the mechanism of RTA dh404 in GBM8401 and U87MG cells, the expression levels of apoptosis-, autophagy-, and cell cycle-associated proteins were individually detected using Western blot analysis (Figure 6). We measured the relative intensities of apoptosis-associated antibodies, such as caspase-3, PARP, BCL2, and Bax; the relative intensities of autophagy-associated antibodies, such as p62/SQSTM1, LC3BI, and LC3BII; and the relative intensities of cell cycle G2/M arrest-associated antibodies, such as cyclinB1, CDK1, WEE1, p21, and NRF2. As shown in Figure 6, the relative intensities of G2/M-associated proteins cyclinB1, CDK1, and WEE1 were significantly downregulated upon RTA dh404 treatment in GBM8401 cells. Conversely, RTA dh404 increased the expression of cyclin B1, CDK1, and WEE1 proteins in U87MG cells. In addition, p21, a cyclic-dependent kinase 1 inhibitor, was significantly upregulated in RTA dh404-treated GBM8401cells, and RTA dh404 significantly downregulated p21 protein expression in U87MG cells. The relative intensities of apoptosis-associated proteins caspase-3 and cleaved-PARP were significantly upregulated upon RTA dh404 treatment in GBM8401 and U87MG cells. BCL2 expression was significantly downregulated in RTA dh404-treated GBM8401 and U87MG cells. Finally, the relative intensities of autophagy-associated proteins, such as p62/SQSTM1, LC3BI, and LC3BII, were significantly upregulated upon RTA dh404 treatment in GBM8401 and U87MG cells

### 2.6. RTA dh404 Modulated the Expression of Apoptosis-, Autophagy-, and G2/M Cell Cycle-Associated Genes in Glioma Cells

#### 2.6.1. Top 20 KEGG Pathway Analyses Were Performed after Gene Ontology Cluster Analysis

To systematically understand the central pathways involved in the synthetic triterpenoid RTA dh404, we performed KEGG (Kyoto Encyclopedia of Genes and Genomes) pathway enrichment analysis of GBM8401 and U87MG cells treated with 8 µM RTA dh404. The results revealed the top 20 significantly enriched KEGG pathways in GBM8401 and U87MG cells. As shown in Figure 7A, there were significant differences following GBM8401 treatment with RTA dh404 in genes associated with the cell cycle, autophagy, cellular senescence, amyotrophic lateral sclerosis, Shigellosis, DNA replication, Human T-cell leukemiavirus-1 infection, neurodegeneration pathway, Huntington’s disease, mitophagy, protein processing in the endoplasmic reticulum, small cell lung cancer, prion disease, lysine degradation, endocytosis, Kaposi sarcoma-associated herpes virus infection, Valine, Leucine, isoleucine degradation, nucleocytoplasmic transport, and thermogenesis. As shown in Figure 7B, there were significant differences following the treatment of U87MG with RTA dh404 in genes regulating the MAPK signaling pathway, FoxO signaling pathway, viral carcinogenesis, cellular senescence, human papillovirus infection, p53 signaling pathway, Notch signaling pathway, epithelial cell signaling in Helicobacter Pylori infection, protein processing in endoplasmic reticulum, autophagy, TNF signaling pathway, prostate cancer, small cell lung cancer, AGE-RAGE signaling pathway in diabetic complications, Epstein–Barr virus infection, focal adhesion, cell cycle, TGF-beta signaling pathway, human T-cell leukemiavirus-1 infection, and Hippo signaling pathway. When infection-, degenerative disease-, and specific cancer-related genes were excluded, we found that treatment of GBM8401 and U87MG with RTA dh404 modulated apoptosis-, autophagy-, and cell cycle-related signaling pathways, suggesting that RTA dh404 activates or affects apoptosis, autophagy, and cell cycle signaling pathways.

#### 2.6.2. Heatmap Clustering Analysis of Differentially Expressed Genes following GBM8401 and U87MG Cells Treatment with CDDO-TFEA and RTA dh404

A heatmap cluster analysis (as Figure 8) identified eleven apoptosis-related genes (*CASP3, CASP9, CASP8, Bax, BCL-XL, FADD, TRADD, Apaf-1, PARP1, NOXA*, and *Cytc*). In addition, eight autophagy-related genes (*SQSTM1* (p62), *MAP1LC3B* (LC3B), *BECN1, mTOR, JNK1, ATG5, ATG7,* and *ATG12*) were identified. Finally, eighteen cell cycle-related genes (*ANAPC2, CCND1, CCNE1, CDK4, CCNG1, CCNH1, CDK7, CDK2, CCNA1, CCNB1, CDK1, WEE1, CDC25A, CAD25B, CDC25C, GADD45A, GADD45B*, and *GADD45G*) were identified. These findings suggested that RTA dh404 treatment affects apoptosis, autophagy, and cell cycle-related gene regulation.

#### 2.6.3. Verification of RTA dh404 Treated GBM8401 and U87MG mRNA Results via NGS Analysis

We confirmed that RTA dh404 induces apoptosis-, autophagy-, and cell cycle-related mRNA gene regulation in glioma cells (Figure 9). We divided the expression levels in the RTA dh404-treated group by those of the vehicle-treated group and considered genes with higher than 2-fold changes to be substantially upregulated and those with less than 2-fold changes to be downregulated. NGS analysis identified 36 genes related to apoptosis (11), autophagy (8), and the cell cycle (18) significantly altered in GBM8401 and U87MG cells treated with RTA dh404 demonstrated by a Z value higher than that of the blank control group. In a heatmap cluster analysis of the eleven apoptosis-related TPM genes, three genes (*Apaf-1, Bcl-2,* and *BCL-XL*) in GBM8401 cells and three genes (*NOXA, BCL-XL*, and *FADD*) in U87MG cells treated with RTA dh404 showed a Z value was higher than that of the blank control group. In addition, of the eight autophagy-related genes, four genes (*SQSTM1* (p62), *MAP1LC3B* (LC3B), *ATG5*, and *ATG12*) in GBM8401 cells and two genes (*SQSTM1* (p62) and *MAP1LC3B* (LC3B)) in U87MG cells treated with RTA dh404 showed a Z value was higher than that of the blank control group. Furthermore, of the eighteen cell-cycle-related genes, eight genes (*CCNE1, CCNA1, GADD45A, GADD45B, GADD45G, CCNH1, CDK2*, and *CDK1*) in GBM8401 cells and eight genes (*CCNE1, CDK7, CCNA1, GADD45A, GADD45B, GADD45G, ANAPC2,* and *CCND1*) in U87MG cells treated with RTA dh404 showed a higher Z value than that of the blank control group.

## 3. Discussion

Synthetic compounds derived from purified plant-derived products play a pivotal role in drug research and development owing to their structural diversity and pleiotropic properties. RTA dh404 is a synthetic oleanolic acid compound with many biological properties including anti-inflammatory, antioxidant, and neuroprotective effects. However, the mechanism underlying the action of this bioactive compound remains unclear. In this study, we found that RTA dh404 attenuated the viability of GBM8401 and U87MG cells. Our findings demonstrate that RTA dh404 induces caspase-dependent apoptosis, autophagy, and G2/M cell cycle arrest in GBM8401 and U87MG cells. These effects may involve multiple signaling pathways, including cell cycle, autophagy, MAPK, FoxO, p53, and Notch signaling pathways.

### 3.1. Apoptosis Is a Programmed Cell Death and Homeostatic Mechanism to Maintain Cell Populations, as Well as a Defense Mechanism in the Presence of Toxic Substances

Apoptosis can be triggered by a variety of cellular signals, including intracellular signals, in response to cellular stress or extrinsic factors [17]. Accumulating evidence has indicated that two main signaling pathways participate in the regulation of apoptosis [18,19]: An intrinsic pathway characterized by mitochondrial outer membrane permeabilization and mitochondrial cytochrome c release [20] and an extrinsic pathway initiated by death receptor stimulation [21]. There is an overlap between these pathways, which ultimately leads to the recruitment and activation of caspases [22]. Several studies have provided extensive evidence that synthetic oleanane derivatives inhibit proliferation and induce apoptosis in various cancer cells [8]. In this study, the viability of GBM8401 and U87MG cells was significantly decreased following RTA dh404 treatment in a dose-dependent manner. As shown in Figure 3, the results of the Annexin-FITC/PI assay showed that the percentage of apoptotic cells was significantly increased in the treated cells, and RTA dh404 induces glioblastoma apoptosis. Moreover, the number of active caspase-3 positive cells increased linearly as the concentration of RTA dh404 increased. A significant increase in caspase-3 activity was detected in RTA dh404-treated cancer cells. Therefore, based on RTA dh404′s effects upregulating caspase-3 and inducing apoptosis, we suggest that RTA dh404 causes apoptosis by activating the caspase pathway, especially with high doses of RTA dh404. The pharmacological properties of high-dose cytotoxicity in tumor cells were similar to those of other synthetic oleanane triterpenoids.

### 3.2. Cell Cycle Dysregulation Is a Hallmark of Tumor Cells

The cell cycle is regulated by cyclins, cyclin-dependent kinases (CDKs), and CDK inhibitors (CKIs). The cyclins and CDKs involved in each phase of the cell cycle are different. For example, in the G2/M phase, CDK1 forms a complex with cyclin B1 to promote transition and mitosis, whereas, in the S phase, CDK2 and cyclin A complexes promote DNA synthesis [23]. CKIs, such as p21, inhibit CDK-cyclin activity and prevent cell cycle progression. The G2 checkpoint prevents cells from entering mitosis when DNA is damaged, providing an opportunity to repair and stop the proliferation of damaged cells. Because the G2 checkpoint helps maintain genomic stability, it plays an important molecular role in cancer [24,25]. According to the cell cycle analysis results, exposure to RTA dh404 resulted in a dose-dependent increase in the number of cells in the G2/M phase, suggesting a reduction in mitosis in GBM8401 and U87MG cells. The results indicated that RTA dh404 inhibited GBM8401 and U87MG cell proliferation, which was associated with the cell cycle arrest in the G2/M phase. In addition, Western blot analysis showed that the relative intensities of G2/M-associated proteins cyclin B1, CDK1, and WEE1 were significantly downregulated upon RTA dh404 treatment in GBM8401 cells. RTA dh404 increased the expression of Cyclin B1, CDK1, and WEE1 proteins in U87MG cells. Furthermore, p21, a CKI, was significantly upregulated in RTA dh404-treated GBM8401cells, while RTA dh404 significantly downregulated p21 protein expression in U87MG cells. These results suggest that metformin could induce G2/M phase arrest in GBM8401 and U87MG cells. However, the underlying mechanism in this situation still remains elusive and needs to be further explored.

### 3.3. Autophagy Is a Conserved Cell Survival Pathway in Eukaryotes That Maintains Cell Homeostasis and Responds to Certain Environmental Stressors [26]

It involves the selective degradation of cellular components, leading to the recycling of nutrients and energy production. Autophagy is initiated under stressful conditions caused by the accumulation of damaged organelles and high bioenergy demand [27]. Accumulating evidence has shown a close relationship between autophagy and cancer. However, developing anticancer drugs targeting autophagy has been very difficult, owing to the complex dual roles of autophagy in tumor survival and cell death [28]. Previous studies demonstrated that synthetic oleanane triterpenoids induce autophagy in different cancer cell types [29] including chronic myeloid leukemia and esophageal squamous cancer cells. Samudio et al. demonstrated that CDDO-Me can induce autophagy in chronic myeloid leukemia cells, and Wang et al. demonstrated that CDDO-Me triggers autophagy initiation in esophageal squamous cancer cells. Synthetic oleanane triterpenoids were confirmed to form extensive double-membrane vesicles, and immunohistochemistry also revealed increased cytoplasmic staining of LC3B, a protein commonly associated with autophagosomes [30]. In addition, synthetic oleanane triterpenoids enhance the conversion of LC3 from the cytosolic form (LC3-I) to the proteolytic and lipidated form (LC3-II) and upregulate beclin 1 [31]. In the present study, the relative intensities of autophagy-associated proteins, such as p62/SQSTM1, LC3BI, and LC3BII, were significantly upregulated in GBM8401 and U87MG cells upon RTA dh404 treatment. Taken together, these results indicate that synthetic oleanane triterpenoids induce autophagy in cancer cells. This feature could be used as a chemopreventive agent for cancer cells.

### 3.4. The Anticancer Strategy of Chemotherapeutic Drugs Involves the Induction of Apoptosis, Autophagy, and Cell Cycle Arrest in Cancer Cells

NGS analysis showed that RTA dh404 induced apoptosis and autophagy in glioblastoma cells and arrested cells in the G2/M phase. First, RTA dh404 promoted or inhibited genes associated with cancer cell apoptosis, including three genes (Apaf-1, Bcl-2, and BCL-XL) in GBM8401 cells and three genes (NOXA, BCL-XL, and FADD) in U87MG cells. In addition, RTA dh404 promoted or repressed genes related to autophagy in cancer cells, including four genes (SQSTM1(p62), MAP1LC3B(LC3B), ATG5, and ATG12) in GBM8401 cells and two genes (SQSTM1 (p62) and MAP1LC3B (LC3B)) in U87MG cells. Finally, RTA dh404 promoted or repressed genes related to the cell cycle, including eight genes (CCNE1, CCNA1, GADD45A, GADD45B, GADD45G, CCNH1, CDK2, and CDK1) in GBM8401 cells and eight genes (CCNE1, CDK7, CCNA1, GADD45A, GADD45B, GADD45G, ANAPC2, and CCND1) in U87MG cells. In view of the above results, we believe that RTA dh404 is a potential anticancer drug against glioblastoma that can induce cell cycle arrest and allow cell death through autophagy and apoptosis, thereby achieving anticancer effects.

To summarize the results of this article, RTA dh404 can induce glioma cell cycle G2/M phase arrest and induce apoptosis and autophagy at high concentrations; these two strategies are pathways to inhibit cancerous tumors (Figure 10). High concentrations of RTA dh404 inhibit cell proliferation and induce caspase-3-associated apoptosis. In addition, RTA dh404 also induces cell cycle arrest by mediating cell-cycle-related regulatory proteins, thereby enhancing the G2 checkpoint regulatory function, leading to cell aggregation in the G2/M phase. Therefore, RTA dh404 is an anticancer drug with dual effects of inducing its own apoptosis and inducing cell cycle arrest.

## 4. Materials and Methods

### 4.1. Reagents and Chemicals

The chemicals and reagents used in this study were (1) RTA dh404, which was purchased from Cayman Chemicals; (2) PrestoBlue™ Cell Viability Reagent from ThermoFisher/Invitrogen; (3) fetal bovine serum (FBS), the antibiotics penicillin/streptomycin (P/S), and modified Eagle medium (MEM) were purchased from Gibco and Roswell Park Memorial Institute (RPMI) 1640 medium (USA); (4) phosphate-buffered saline (PBS), dimethyl sulfoxide (DMSO), trypsin-EDTA (0.25%), and Trypan Blue Solution were purchased from Sigma (St Louis, MO); (5) the polyvinylidene fluoride membrane (PVDF) (Millipore) and molecular weight markers were purchased from Bio Rad (USA); and (6) propidium iodide (PI) (USA).

Western blot antibodies were also purchased from commercial vendors and used at the indicated dilutions: Cyclin B (1:1000; Proteintech; 55004-1-AP), CDK1 (1:1000; Cell Signaling; E1Z6R), Wee1 (1:1000; Proteintech 14375-1-AP), p21 (1:1000; Cell Signaling E2R7A); Bcl-2 (1:1000), Nrf2 (1:1000; Cell Signaling; D1Z9C), Bcl-2(1;1000; Proteintech), Bax (1:1000; Proteintech), Caspase-3 (1:1000; Affinity), PARP (1:1000; Affinity), LC3B (1:1000; Invitrogen), p62/SQSTM1 (1:1000; Affinity), and β-actin (1:20000; Sigma; A5441).

### 4.2. Cell Culture

Human brain malignant glioma GBM8401 cells (glioblastomas) were purchased from the Food Industry Research and Development Institute (Hsinchu, Taiwan). Human brain malignant glioma U87MG cells (astrocytoma) were obtained from the Bioresource Collection and Research Center (BCRC, Hsinchu, Taiwan). U87MG cells were cultured in MEM supplemented with 10% FBS and 1% P/S. GBM8401 cells were cultured in RPMI medium supplemented with 10% FBS and 1% P/S. Both cell lines were incubated at 37 °C and 5% CO_2_.

### 4.3. Cell Viability Assay

Cell viability was tested using the PrestoBlue™ assay, as described by Mossman (1983) [32]. The GBM8401 and U87MG cells were seeded into a 96-well culture plate at 3000 cells/well incubated in an atmosphere containing 5% CO_2_, saturated humidity, and 37 °C for 24 h. The next day, cells were exposed to various concentrations of RTA dh404 (0, 2, 4, or 8 µM) or DMSO as a vehicle control for 24–72 h. Cells in each well were treated with the PrestoBlue reagent for at least 10 min. Finally, absorbance was measured at 560 nm (OD560) and 590 nm (OD590) using a multiwell plate reader (BioTek, Taipei, Taiwan). IC50 values were determined. All samples were assayed in triplicate, and the mean values were calculated for each experiment. The results are expressed as a percentage of the control, which was considered to be 100%. All assay results are expressed as the mean ± standard error of the mean (SEM).

### 4.4. Colony-Forming Assay

First, 1 × 10^5^ GBM8401, and U87MG cells treated with RTA dh404 (0, 2, 4, or 8 µM) were seeded in a six-well plate and incubated at 37 °C in a 5% CO_2_ atmosphere for 14 days; the medium was changed every 72 h. Cells were stained with crystal violet and observed under a microscope.

### 4.5. Cell Cycle Analysis

Cell cycle analysis was used to analyze the effect of drug treatment on the cell cycle and was performed as described previously [33]. To assess cell cycle progression, cells were seeded into culture dishes at 3 × 10^5^ cells per dish, incubated for 24 h, and then exposed to different concentrations of RTA dh404 (0, 2, 4, or 8 µM) for 24 h. Next, cells were harvested and fixed in 3 mL of 70% ethanol for at least 8 h at −20 °C. The cells were then removed with ethanol and washed at least once with phosphate-buffered saline (PBS). Cell DNA was stained with a PI/Triton X-100/ RNase A solution for 30 min. Fluorescence was measured using a FACSCalibur flow cytometer (BD, USA), and the data were analyzed using WinMDI 2.9 free software (BD, USA). The results are expressed as a percentage of the control, which was considered to be 100%. All assay results are expressed as the mean ± SEM.

### 4.6. Assessment of Apoptosis

The Annexin V-FITC Apoptosis Detection Kit I was used to determine whether apoptosis was involved in the inhibitory effect of RTA dh404 on cell viability in GBM8401 and U87MG cells. Annexin V-FITC Apoptosis Detection Kit I was used according to the manufacturer’s instructions, as described previously [34,35]. GBM8401 and U87MG cells were cultured in 6-well culture plates (Orange Scientific, EU). RTA dh404 (0, 2, 4, or 8 µM) was used to treat the GBM8401 and U87MG cells for 24 h. The cell suspension was centrifuged at 1000 rpm for 3 min and the supernatant was removed. The cells were washed with PBS and centrifuged at 1000 rpm for 3 min. After removing the supernatant, the treated cells were added to Annexin V-FITC and PI (100 mg/mL) in 1 × Annexin binding buffer and protected from light for 15 min at room temperature. Then, 400 μL of the 10-fold diluted Annexin V Binding Solution was added, the solution was prepared for the flow cytometric assay using Falcon tip centrifuge tubes, analyzed using FACScan flow cytometry (FACSCalibur, BD Pharmingen, USA), and the data were analyzed using WinMDI 2.9 free software (BD, USA).

### 4.7. Caspase-3 Activity Assay

The activity of caspase-3 was detected using a Caspase-3 (activity) FITC staining kit according to the manufacturer’s instructions [36,37]. GBM8401 and U87MG cells were seeded in 6-well plates (1 × 10^6^/mL) and incubated overnight. Cells were treated with different concentrations of RTA dh404 (0, 2, 4, or 8 µM) for 24 h. After 24 h, the required amount of BD Perm/Wash™ buffer (10X) was diluted 1:10 in distilled water. The cells were washed twice with cold 1X PBS and then resuspended in BD Cytofix/Cytoperm™ solution. Next, the cells were stained with the FITC rabbit anti-active caspase 3 antibody (FACSCalibur, BD Pharmingen) and incubated for 30 min. After centrifugation at 2500 rpm for 5 min, the supernatant was removed, the cells were washed with PBS, resuspended in 0.5 mL PBS, and centrifuged again. Finally, the cells were transferred from centrifuge tubes to Falcon centrifuge tubes, the samples were then analyzed using FACScan flow cytometry (FACSCalibur, BD, USA), and the data were analyzed using WinMDI 2.9 free software (BD Pharmingen, USA).

### 4.8. Western Blotting

Western blotting was performed as previously described [38,39]. All cell lysates (50–80 µg) were prepared in an ice-cold lysis buffer. Protein samples were loaded onto 10–12% sodium dodecyl sulfate (SDS)-polyacrylamide gel electrophoresis membranes for electrophoretic separation and then transferred to PVDF membranes (Millipore) at 500 mA for 2 h. After blocking the buffer overnight with 5% non-fat dry milk in PBS containing Tween20 (PBST), the membranes were incubated with primary antibodies [Cyclin B1 (1:1000; Proteintech; 55004-1-AP), CDK1 (1:1000; Cell Signaling; E1Z6R), NRF2 (1:1000; Proteintech; 16396-1-AP), and β-actin (1:20,000; Sigma; A5441)] for 2 h at room temperature or overnight at 4 °C. Membranes were then washed once with PBST and twice with PBS, incubated with the secondary antibody Li-COR (Taipei, Taiwan) at a 1:20,000 dilution for 30–40 min, and washed again. Antigens were visualized using a near-infrared fluorescence imaging system (Odyssey LICOR, USA), and these data were interpreted using Odyssey2.1 software or a chemiluminescence detection kit (ECL; Amersham Corp., Arlington Heights, IL, USA). Densitometry analysis (including the integrated density of bands) was carried out using ImageJ (NIH), followed by normalization of the measured values to β-actin as a loading control.

### 4.9. Next-Generation Sequencing (NGS)

Following 24 h of incubation, total RNA was extracted using the RNAzol RT reagent. Taking 1 µg of extracted total RNA and purified using oligo (dT) as a starting material, magnetic beads were used to capture eukaryotic mRNA and fragment the mRNA. Using fragmented mRNA as a template, random primers and reverse transcriptase were used to synthesize the first strand of cDNA, followed by the addition of the reagents: dNTPs, RNase H, and DNA polymerase, to synthesize double-stranded cDNA products. Double-stranded cDNA purified by AMPure XP (Beckman Coulter, Beverly, USA) magnetic beads, which were then subjected to end repair and 3′ adenylation, sequenced adapters were attached, and the library was completed after the amplification products were purified by PCR polymerization and magnetic beads. Sequencing of the library was performed using the Agilent BioAnalyzer2100 system for fragment size detection, a Real-Time PCR system was used for library concentration determination, and size 150 PE sequencing was performed on an Illumina NovaSeq 6000 sequencer.

### 4.10. Data Analysis

Data are expressed as mean ± SEM of at least three independent experiments. One-way and two-way analyses of variance with GraphPad Prism’s test were used for statistical analysis. Statistical significance was set at a *p* value < 0.05. All statistical analyses were performed using SPSS software (version 24.0; SPSS Inc., Chicago, IL, USA).

## 5. Conclusions

In conclusion, RTA dh404, a novel synthetic oleanolic acid derivative, is a potential anticancer agent against human glioma cells in vitro. Our findings indicated that RTA dh404 suppressed cell viability, induced caspase-dependent apoptosis and autophagy, and triggered G2/M cell cycle arrest in U87MG and GBM8401 cells. RTA dh404 induces cell cycle arrest and cellTable death through autophagy and apoptosis, thereby achieving anticancer effects. However, the effect of RTA dh404 in vivo and its role remain to be further studied.

## Figures and Tables

**Figure 1 ijms-24-04006-f001:**
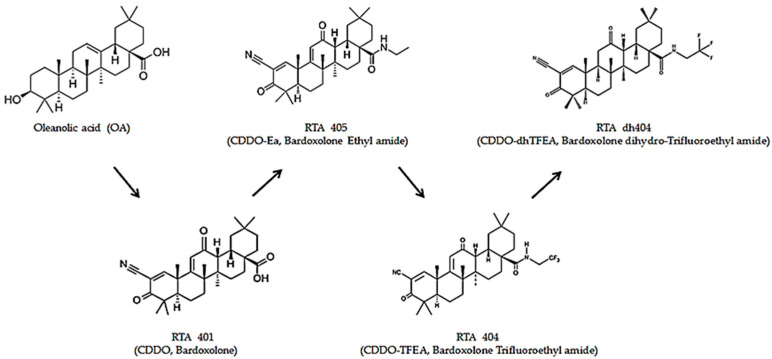
Structures of the synthetic oleanane triterpenoids. OA, Oleanolic Acid; RTA 401, CDDO, Bardoxolone; RTA 405, CDDO-Ea, Bardoxolone Ethyl amide; RTA 404, CDDO-TFEA, Bardoxolone Trifluoethyl Amide; RTA dh404 CDDO-dhTFEA, Bardoxolone dihydro-Trifluoroethyl Amide.

**Figure 2 ijms-24-04006-f002:**
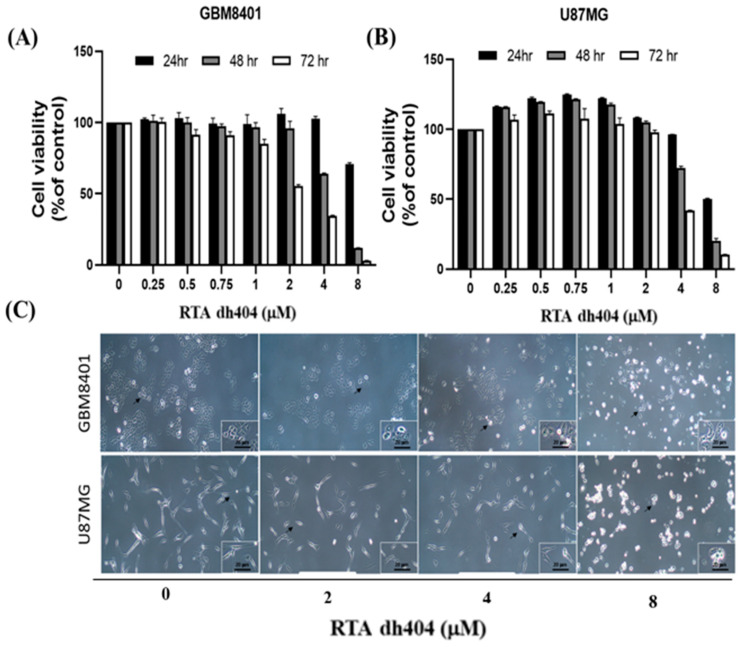
RTA dh404 inhibited the viability of GBM8401 and U87MG cells in a time- and dose-dependent manner. Cell viability of RTA dh404-treated glioma cells between 24 to 72 h. (**A**) GBM8401 and (**B**) U87MG cells were treated with different concentrations of RTA dh404 (0, 2, 4, or 8 µM) for 24–72 h. The control groups were exposed to (DMSO). Cell viability was measured using PrestoBlue™ reagent assay. (**C**) The morphology of GBM8401 and U87MG cells after treatment with RTA dh404 (0, 2, 4, or 8 µM) by 24h. (**D**) Colony-formation assays were performed using GBM8401 and U87MG cells after treatment with RTA dh404 (0, 2, 4, and 8 µM). The results are shown as mean ± SD of three independent experiments. ** *p* < 0.001 and *** *p* < 0.0001 compared with control.

**Figure 3 ijms-24-04006-f003:**
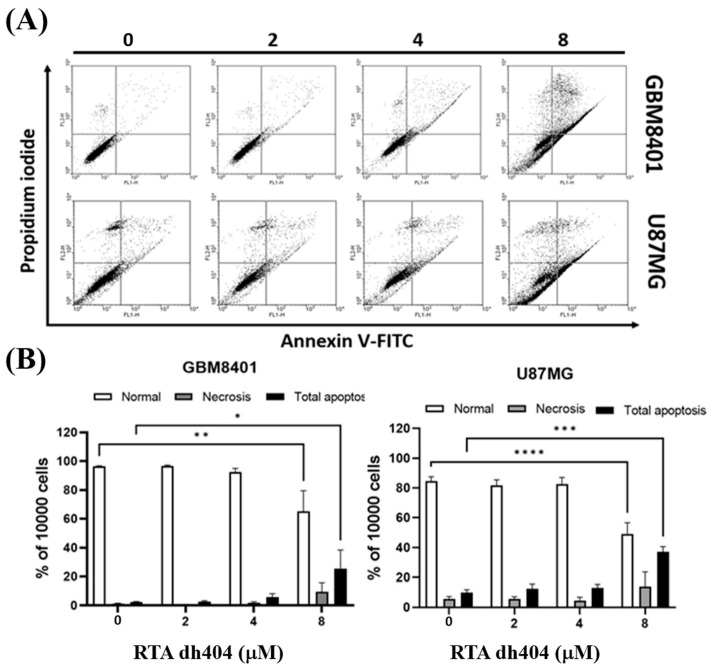
High-dose RTA dh404 induced apoptosis in glioma cells. (**A**,**B**) GBM8401 and U87MG were treated with RTA dh404 (0, 2, 4, or 8 µM) for 24 h. Cells were harvested and stained with Annexin V-FITC and PI, and apoptotic cell death was analyzed using flow cytometry. * *p* < 0.05, ** *p* < 0.001, *** *p* < 0.0001, and **** *p* < 0.00001 compared with control.

**Figure 4 ijms-24-04006-f004:**
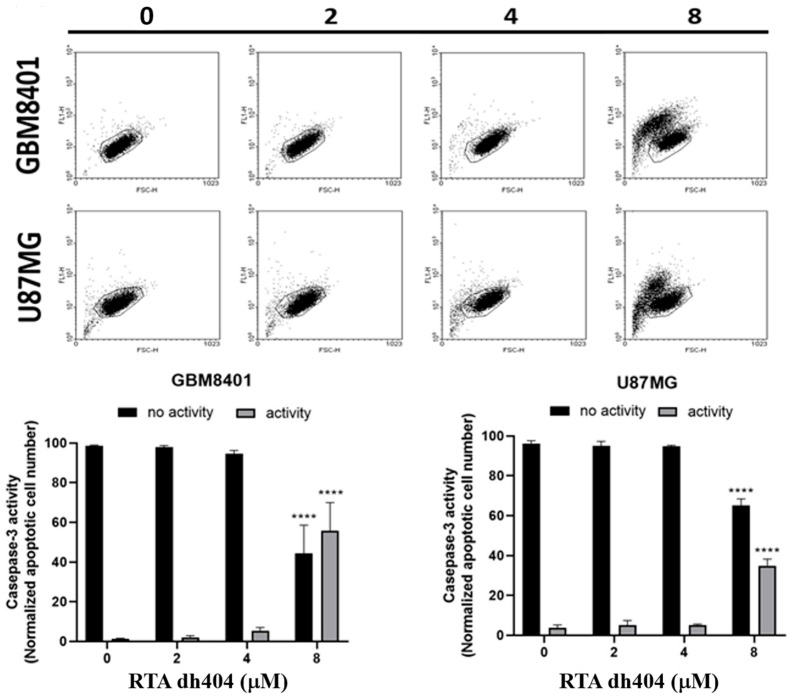
High-dose RTA dh404 treatment induces caspase-3 activation in glioma cells. GBM8401 and U87MG were treated with different concentrations of RTA dh404 (0, 2, 4, or 8 µM) for 24 h. Cells were harvested, stained with caspase 3-FITC, and activated caspase-3 was analyzed using flow cytometry. ***** p* < 0.00001 compared with control.

**Figure 5 ijms-24-04006-f005:**
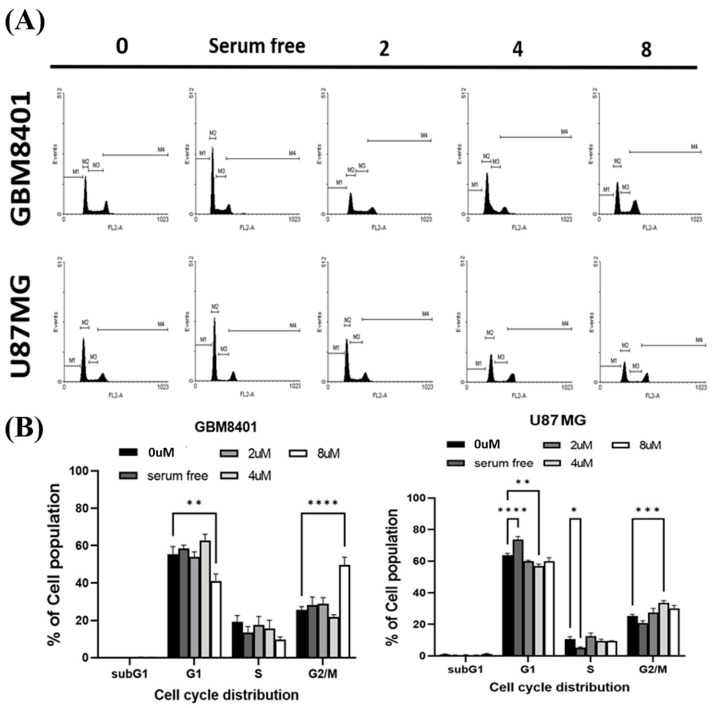
RTA dh404 induced the accumulation of G2/M cell-cycle-arrested glioma cells. RTA dh404 increased cell cycle G2/M phase glioma cell distribution. (**A**) Cell cycle analysis of GBM8401 and U87MG cells treated with different concentrations of RTA dh404 (0, 2, 4, or 8 µM) for 24 h. (**B**) These results showed that RTA dh404 increased the percentage of the cell population in the G2/M phase. Cells were stained with PI by flow cytometry. A serum-free control group did not include 10% FBS. The results are shown as mean ± SD of three independent experiments. * *p* < 0.05, ** *p* < 0.001, *** *p* < 0.0001, and ***** p* < 0.00001 compared with control.

**Figure 6 ijms-24-04006-f006:**
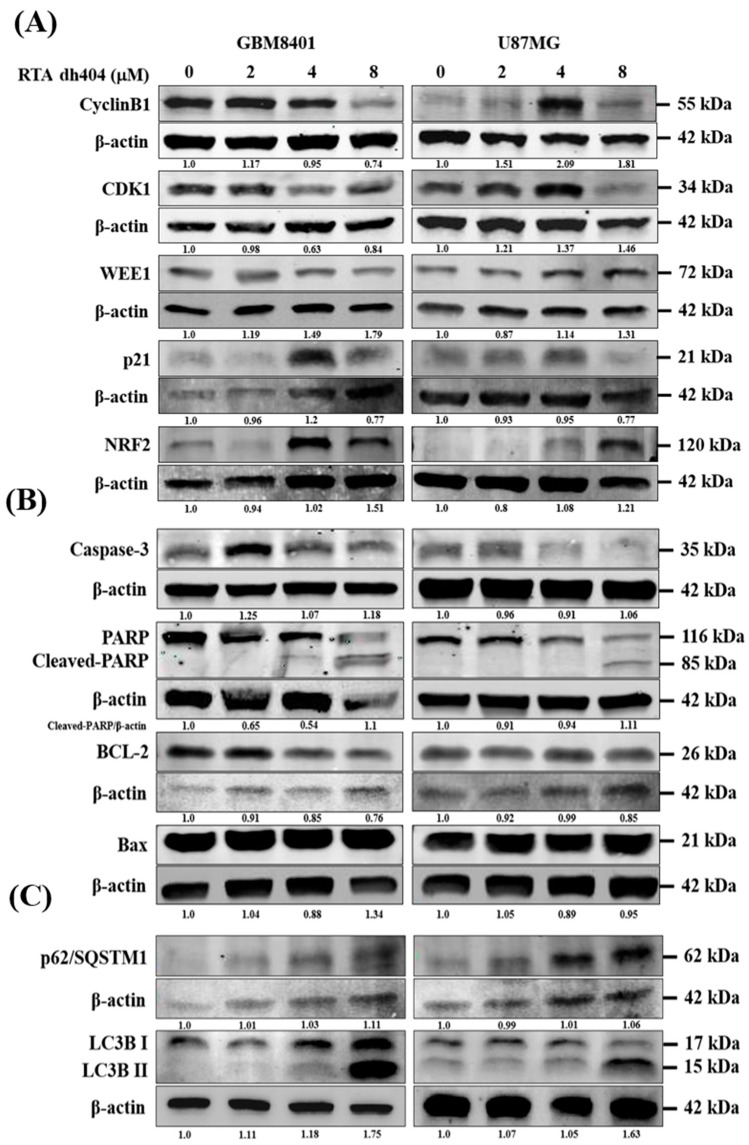
RTA dh404 regulates the expression of cell cycle G2/M-, apoptosis-, and autophagy-associated proteins in glioma cells. (**A**–**C**) GBM8401 and U87MG were treated with RTA dh404 (0, 2, 4, or 8 µM) for 24 h, and cell lysates were separated by SDS-PAGE and analyzed by western blotting using cell cycle, apoptosis, and autophagy-related antibodies. β-actin was used as a loading control.

**Figure 7 ijms-24-04006-f007:**
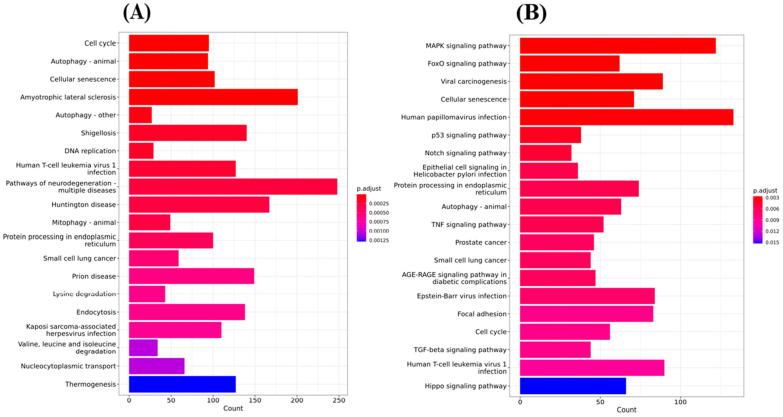
Top 20 KEGG pathway analyses were performed after Gene Ontology cluster analysis. (**A**) Gene-based important KEGG pathway analysis on GBM8401 cells treated with RTA dh404. (**B**) Gene-based important KEGG pathway analysis on U87MG cells treated with RTA dh404.

**Figure 8 ijms-24-04006-f008:**
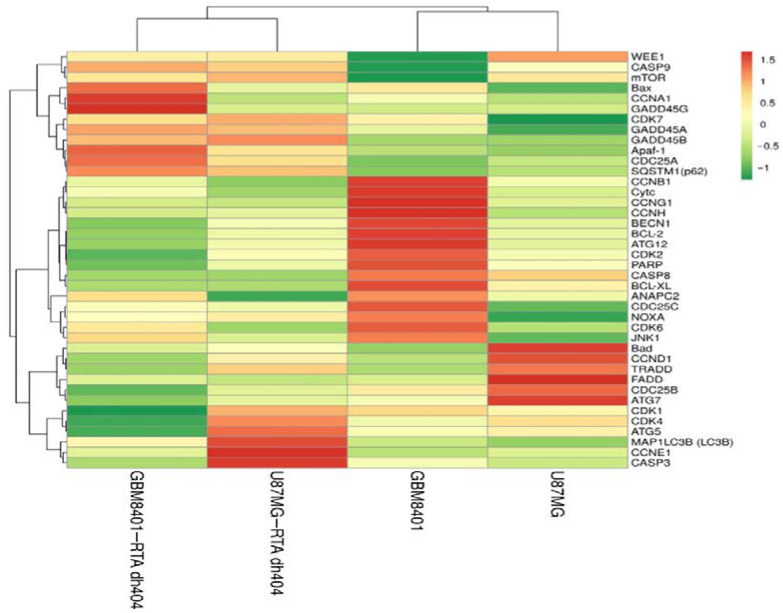
Heatmap clustering analysis of differentially expressed genes following GBM8401 and U87MG cells treatment with CDDO-TFEA and RTA dh404. GBM8401 cells were treated with CDDO-TFEA or RTA dh404 and the transcripts per million (TPM) gene expression was compared to blank. U87MG cells were treated with CDDO-TFEA or RTA dh404 and the TPM gene expression was compared to blank. Red indicates an increase in gene expression after treatment, while green indicates a decrease. Genes that change consistently after treatment with compounds are highlighted in red and green boxes.

**Figure 9 ijms-24-04006-f009:**
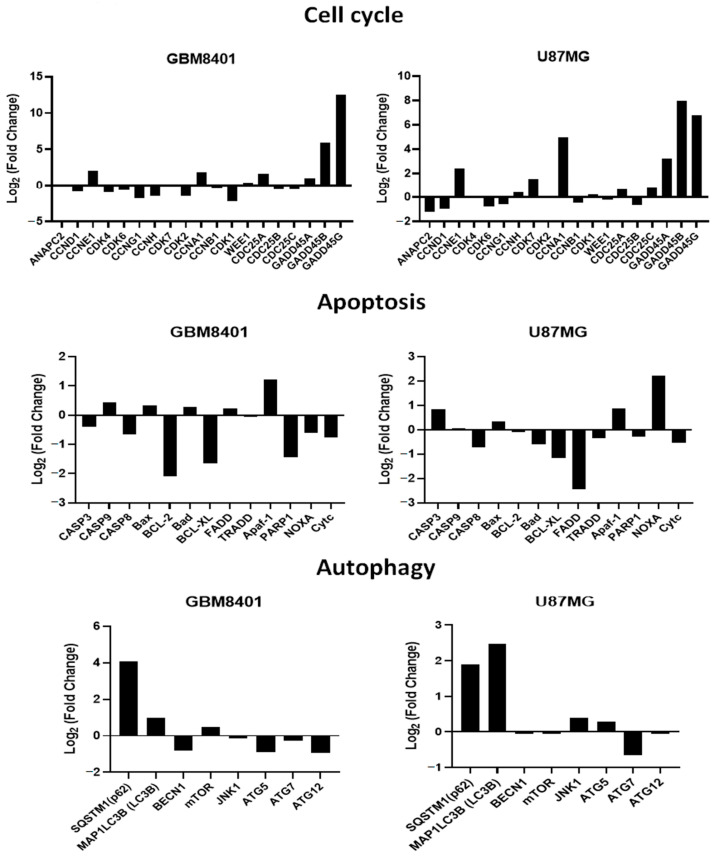
RTA dh404 regulated cell cycle, apoptosis, and autophagy-associated mRNA in GBM8401 and U87MG cells. The genes downregulated or upregulated in GBM8401 and U87MG cells following exposure to RTA dh404. Cell cycle G2/M phase-related gene log fold change (log Fc) expression profiles were studied in GBM8401 and U87MG cells exposed for 24 h to 8 µM CDDO-TFEA or RTA dh404.

**Figure 10 ijms-24-04006-f010:**
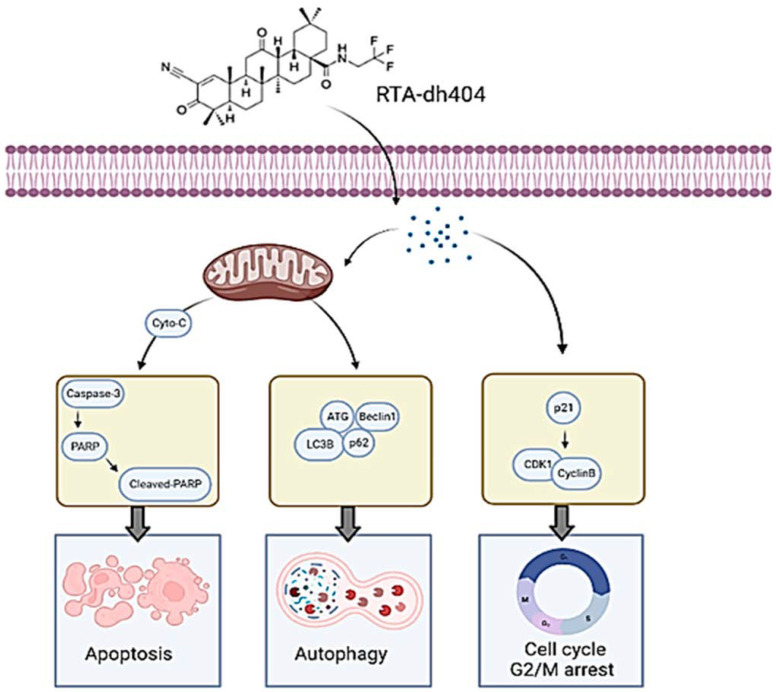
RTA dh404 induces apoptosis, autophagy, and G2/M cell cycle arrest in Glioblastoma Multiforme. RTA dh404 induces G2/M phase cell cycle arrest by regulating the p21, cyclin B, and cdk1 protein; and a high-dose of RTA dh404 induces apoptosis and autophagy in Glioblastoma Multiforme. This figure was created using BioRender.com.

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
