# Peer review of "RTA dh404 Induces Cell Cycle Arrest, Apoptosis, and Autophagy in Glioblastoma Cells"

_ijms, 2023, doi:10.3390/ijms24044006_

Round 1

Reviewer 1 Report

In this research, RTA-dh404 has been promoted as a potential drug candidate for the treatment of GBM, since it can induce cell cycle arrest and apoptosis. Overall, the manuscript is in good shape. Yet there are some concerns need to be addressed before the acceptance for publication.  

1. Need references to support RTA-dh404 can activate Nrf2 and inhibit NF-Kb.

2. For the cell viability experiment, more concentrations (i.e. 0, 0.01, 0.05, 0.1, 0.5, 1, 5, 10 uM) of the tested compound should be included to draw the inhibition curve, which may therefore help to calculate the IC50. In the abstract, it indicates cell viability was evaluated by MTT, yet in the result it stated as the PrestoBlue assay, please check and keep consistent.

3. Figure 2C, showing the data of the morphological change is fine, but it is not quantitative. Colony formation assay would be a better option. 

4. Figure 3, in the case for the differentiation of necrosis and apoptosis, the researchers should include more markers (i.e. TNFa, IL1b). Otherwise, there are necrosis/apoptosis kits available to better illustrate this statement. 

5. In Figure 2, RTA-dh404 could affect cell viability in a dose dependent manner. However, in figure 6, the dose dependent effect in not fully represented. 

6. Including animal studies would make this research more convincing. 

Author Response

Department of Neurosurgery

Kaohsiung Medical University Hospital

Kaohsiung, Taiwan

January 19th, 2023

Dear Reviewer 1:

Thank you for giving me the opportunity to submit a revised draft of our manuscript which is titled: RTA dh404 Induces Cell Cycle Arrest, Apoptosis, and Autophagy in Glioblastoma Cells. We appreciate the time and effort that you and the reviewers have dedicated to providing your valuable feedback on our manuscript. We are grateful to the reviewers for their insightful comments on this paper. We have been able to incorporate changes to reflect most of the suggestions provided by the reviewers. We have highlighted the modification within the manuscript and point-by-point response to the reviewers’ comments and concerns in this revision.

Comments from Reviewer 1

In this research, RTA-dh404 has been promoted as a potential drug candidate for the treatment of GBM, since it can induce cell cycle arrest and apoptosis. Overall, the manuscript is in good shape. Yet there are some concerns that need to be addressed before the acceptance for publication.

Comment 1: Need references to support RTA-dh404 can activate Nrf2 and inhibit NF-Kb.

Response to comment 1:

Agree. We have, accordingly modified the manuscripts and results to emphasize this point. We have added references to the manuscript to let readers know more about RTA dh404 can activate Nrf2 and inhibit NF-Kb. (As References 8-11; Page line 577 -Page line 586)

References:

  1. Aminzadeh MA, Reisman SA, Vaziri ND, Khazaeli M, Yuan J, Meyer CJ. The synthetic triterpenoid RTA dh404 (CDDO-dhTFEA) restores Nrf2 activity and attenuates oxidative stress, inflammation, and fibrosis in rats with chronic kidney disease. Xenobiotica. 2014 Jun;44(6):570-8. doi: 10.3109/00498254.2013.852705. Epub 2013 Nov 6. PMID: 24195589; PMCID: PMC4046874.
  2. Dinkova-Kostova AT, Liby KT, Stephenson KK, Holtzclaw WD, Gao X, Suh N, Williams C, Risingsong R, Honda T, Gribble GW, Sporn MB, Talalay P. Extremely potent triterpenoid inducers of the phase 2 response: correlations of protection against oxidant and inflammatory stress. Proc Natl Acad Sci U S A. 2005 Mar 22;102(12):4584-9. doi: 10.1073/pnas.0500815102. Epub 2005 Mar 14. PMID: 15767573; PMCID: PMC555528.
  3. Liby KT, Yore MM, Sporn MB. Triterpenoids and rexinoids as multifunctional agents for the prevention and treatment of cancer. Nat Rev Cancer. 2007 May;7(5):357-69. doi: 10.1038/nrc2129. Epub 2007 Apr 19. PMID: 17446857.
  4. Liby KT, Sporn MB. Synthetic oleanane triterpenoids: multifunctional drugs with a broad range of applications for prevention and treatment of chronic disease. Pharmacol Rev. 2012 Oct;64(4):972-1003. doi: 10.1124/pr.111.004846. Epub 2012 Sep 10. PMID: 22966038; PMCID: PMC3462991.

Comment 2: For the cell viability experiment, more concentrations (i.e. 0, 0.01, 0.05, 0.1, 0.5, 1, 5, 10 uM) of the tested compound should be included to draw the inhibition curve, which may therefore help to calculate the IC50. In the abstract, it indicates cell viability was evaluated by MTT, yet in the result, it stated as the PrestoBlue assay, please check and keep it consistent.

Response to comment 2:

Agree. We have accordingly modified the manuscripts and results to emphasize this point. We have used more concentrations (i.e. 0, 0.25, 0.5, 0.75, 1, 2, 4, 8 uM) of the tested RTA-dh404, and IC50 values had been calculated. As shown in Figure 2, the viability of GBM8401 and U87MG cells was significantly decreased following RTA dh404 treatment in a dose-dependent manner at 48 hrs and 72 hrs, however, was significantly decreased high dose (8uM) at 24h without dose-dependent manner (24 h; U87MG; y=-8.158x + 121.83, R2= 0.8196, GBM8401; y=-3.3971x + 104.87, R2= 0.6795); (48 h; U87MG; y=-12.255x + 121.78, R2=0.9157, GBM8401; y=-11.342x + 106.8, R2=0.9674); (72 h; U87MG; y=-13.266x + 112.32, R2=0.9279, GBM8401; y=-12.73x + 96.385, R2=0.9407), indicating that RTA-dh404 dose-dependently inhibited clone formation in GBM8401 and U87MG cells.) administered over 24–72 h. The values of IC50 for RTA-dh404 are 3uM (24 hr), 5.4 uM (48 hr), and 8.6 uM (72 hr) in GBM8401 cells, and 3.3uM (24 hr), 4.5 uM (48 hr), 6 uM (72 hr) in U87MG. (As Figure 2; Page line 95-Page line 96)

And thank the reviewers for reminding us that there are inconsistencies in the experimental methods of cell viability in the abstract, methods, and materials. We have checked and corrected it to PrestoBlue™ reagent. (As Abstract; Page line 19 -Page line 20)

Comment 3: Figure 2C, showing the data of the morphological change is fine, but it is not quantitative. A colony formation assay would be a better option.

Response to comment 3:

Thank you for pointing this out. We agree with this comment. It is true that the morphology after treatment with RTA-dh404 cannot be quantified, so in the quantitative method of cell viability, we added Colony formation assay, and the results have been written in M&M and Results and presented in figure 2D (As Figure 2D; Page line 96-Page line 97)

Comment 4: Figure 3, in the case for the differentiation of necrosis and apoptosis, the researchers should include more markers (i.e. TNFa, IL1b). Otherwise, there are necrosis/apoptosis kits available to better illustrate this statement.

Response to comment 4:

Thank you for this suggestion. It is very important to distinguish between necrosis and apoptosis caused by drugs. Therefore, we analyzed cell death using Annexin V-FITC and PI staining to confirm RTA-dh404 caused apoptosis in glioblastoma cells. More importantly, it was verified that RTA-dh404 would induce apoptosis. Subsequently, a caspase-3 assay was performed to determine whether cell death was triggered by caspase-dependent apoptosis, and some markers such as Caspase3, Bax, BCL-2, PARP, and cleaved-PARP were also used, as RTA-dh404 drug will cause glioma cell apoptosis molecular biological marker. (As Figure 3)

Comment 5: In Figure 2, RTA-dh404 could affect cell viability in a dose-dependent manner. However, in figure 6, the dose-dependent effect is not fully represented.

Response to comment 5:

Thank you for this suggestion. It would have been interesting to explore this aspect. Why can't the dose-dependent effect of WB be fully reflected? Because the cell viability at 24 hours, 48 hours, and 72 hours in the Cell viability experiment is shown in Figure 2, it is found that the cell viability at 48 hours and 72 hours does show a dose-dependent manner, but the cell viability at 24 hours is only statistically significant at high concentrations. There is no dose-dependent manner.

Furthermore, in Western Blotting analysis, 24 hours after the cell line was treated with drugs, the cells were harvested, cell proteins were extracted, and electrophoresis was performed, so no dose-dependent manner as shown in the WB analysis.

Therefore, the above description can explain why cell viability has a dose-dependent manner, while WB analysis does not have a dose-dependent manner. (As Figure 2 and Figure 6)

Comment 6: Including animal studies would make this research more convincing.

Response to comment 6:

Agree. Thank you for this suggestion. It would have been interesting to explore this aspect. The amount of data in this experiment is not small, so we will organize and submit the current research data first, and we have already planned the follow-up animal experiment plan and further research (As Concussion section)

Additional clarifications

In addition to the above comments, all spelling and grammatical errors pointed out by the reviewers have been corrected.

We look forward to hearing from you in due time regarding our submission and to responding to any further questions and comments you may have.

Sincerely,

Prof. Tai-Hsin Tsai

Department of Neurosurgery, Kaohsiung Medical University Hospital

No. 100, Tzyou 1st Road, Sham-min District, Kaohsiung City, Taiwan

Reviewer 2 Report

Reviewer (Remarks to the Author):

The manuscript entitled “RTA-dh404 Induces Cell Cycle Arrest, Apoptosis, and Autophagy in Glioblastoma Cells” by Tai-Hsin Tsai et al. suggestes RTA-dh404 causes G2/M cell cycle arrest and induces apoptosis and autophagy by regulating the expression of cell cycle-, apoptosis-, and autophagy-related genes in human glioblastoma cells. The premise of the work is very interesting, however in its present version, the manuscript requires several significant areas of improvement before consideration for publication.

1) Overall the quality of the microscope data are not particularly strong, often hard to understand. The scale bar information for images should be added to images and Figure Legends, making it difficult for the reader to assess the size between different images.

2) How many independent experiments were done for the western blots. For the western blot loading control protein seems very strong (Figure 6A, 6B, 6C). There is a good chance that the normalization may be done incorrectly? It is better add molecular weights of the protein on the image for all western blots?

3) Were different gels used for these proteins for Figure 6A? The molecular weights of the proteins are very close to each other? The molecular weight of p21 and Caspase 3 is around the 15 kDa bands. How did you separate the bands? Did you run different gels or strip it? If you use different gels, it is better show for all proteins individual beta actins?

4) The manuscript reads as a set of data that doesn't really hang together as nothing is done to draw any strong conclusions. It would be more interesting if authors can provide one separate model Figure showing the results of manuscript.

5) There are some points either discussed haphazardly or overlooked, need to be discussed properly. I wonder if authors would provide a box (containing some bullet points) addressing some major points/ mechanisms/ challenges and/ or answers of some demanding questions of the discussed area.

6) The descriptions of data in the figures also needs to be significantly improved.

7) Although the manuscript shows some interesting correlative data, there are some issues with quality of Figures. The quality of Figures needs to improved.

8) If cell lines were used in the research, a statement addressing the following points must be included in the Materials and Methods section of the manuscript.

b. Whether the cell lines have been tested and authenticated

 c. The method by which the cells were tested

 d. How and when the cells were last tested

If cells were obtained directly from a cell bank that performs cell line characterizations and passaged in the user's laboratory for fewer than 6 months after receipt or resuscitation, re-authorization is not required. In these cases, please include the method of characterization used by the cell bank.

Author Response

Department of Neurosurgery

Kaohsiung Medical University Hospital

Kaohsiung, Taiwan

January 19th, 2023

Dear Reviewer 2:

Thank you for giving me the opportunity to submit a revised draft of our manuscript which is titled: RTA dh404 Induces Cell Cycle Arrest, Apoptosis, and Autophagy in Glioblastoma Cells. We appreciate the time and effort that you and the reviewers have dedicated to providing your valuable feedback on our manuscript. We are grateful to the reviewers for their insightful comments on this paper. We have been able to incorporate changes to reflect most of the suggestions provided by the reviewers. We have highlighted the modification within the manuscript and point-by-point response to the reviewers’ comments and concerns in this revision.

Comments from Reviewer 2:

The manuscript entitled “RTA-dh404 Induces Cell Cycle Arrest, Apoptosis, and Autophagy in Glioblastoma Cells” by Tai-Hsin Tsai et al. suggests RTA-dh404 causes G2/M cell cycle arrest and induces apoptosis and autophagy by regulating the expression of cell cycle-, apoptosis-, and autophagy-related genes in human glioblastoma cells. The premise of the work is very interesting, however, in its present version, the manuscript requires several significant areas of improvement before consideration for publication.

Comment 1: Overall the quality of the microscope data is not particularly strong, and often hard to understand. The scale bar information for images should be added to images and Figure Legends, making it difficult for the reader to assess the size between different images.

Response to comment 1:

 Thanks for pointing this out. We have added the scale bar information for the images to the image and figure legends to improve the quality of the microscopy data, make the images understandable, and allow readers to assess size differences between images. (As Figure 2C)

Comment 2: How many independent experiments were done for the western blots? For the western blot loading control protein seems very strong (Figure 6A, 6B, 6C). There is a good chance that the normalization may be done incorrectly? It is better to add molecular weights of the protein on the image for all western blots?

Response to comment 2:

Thank you for this suggestion. It would have been interesting to explore this aspect. Basically, we performed at least 3+ independent experiments on all western blots. Statistics of at least 3 independent experiments were carried out for Western blot, and charts were drawn, as shown in Figure 6.

Again, for all western blot images, the protein molecular weights are added. Of course, when loading control proteins for Western blots, there is a high chance of incorrect normalization, which should be avoided as much as possible. (As Figure 6)

Comment 3: Were different gels used for these proteins for Figure 6A? The molecular weights of the proteins are very close to each other? The molecular weight of p21 and Caspase 3 is around the 15 kDa bands. How did you separate the bands? Did you run different gels or strip them? If you use different gels, it is better to show all protein's individual beta actins?

Response to comment 3:

Thank you for pointing this out. We agree with this comment. We did at least 3+ independent experiments on all western blots, all western blots used different gels, and we ran different gels, so the bands are independent Yes, are separate bands. Because we use a different gel, as suggested by the reviewer to show the single β-actin for all proteins, as in Figure 6 (As Figure 6)

Comment 4: The manuscript reads as a set of data that doesn't really hang together as nothing is done to draw any strong conclusions. It would be more interesting if the authors can provide one separate model Figure showing the results of the manuscript.

Response to comment 4:

Thank you for this suggestion. Based on the reviewer's suggestion, we provide a separate model diagram to show the results of the manuscript (as shown in the figure), so that the manuscript no longer reads like a set of data, but they really connect together. We have compiled and integrated our experimental data and concluded as follows: RTA dh404 induces caspase-dependent apoptosis, autophagy, and G2/M cell cycle arrest in GBM8401 and U87MG cells. These effects may involve multiple signaling pathways, including cell cycle, autophagy, MAPK, FoxO, p53, and Notch signaling pathways. (As Figure 10)

Comment 5: There are some points either discussed haphazardly or overlooked and need to be discussed properly. I wonder if the authors would provide a box (containing some bullet points) addressing some major points/ mechanisms/ challenges and/ or answers to some demanding questions of the discussed area.

Response to comment 5:

Thank you for this suggestion. The discussion part is one of the areas where we have to improve. We accept the reviewer's suggestion, and we provide the framework of RTA dh404 inducing cell death, including some key points such as apoptosis, autophagy, and G2M cell cycle arrest, etc., and discuss some points in these areas further, rather than casually. In the discussion section,

3.1 RTA dh404 induction of apoptosis.

3.2 RTA dh404 induction of cell cycle arrest in the G2/M Phase.

3.3 RTA dh404 induction of autophagy.

3.4 RTA dh404 induction of cell cycle arrest by the regulation of the expression of the cell cycle, apoptosis, and autophagy-related genes.

(As Figure 10; As Discussion section)

Comment 6: The descriptions of data in the figures also need to be significantly improved.

Response to comment 6:

Thank you for this suggestion. The description of the data in the figure in the article has been rewritten and the description of the data has been improved as much as possible. (As Figure 1-10)

Comment 7: Although the manuscript shows some interesting correlative data, there are some issues with the quality of the Figures. The quality of the Figures needs to improve.

Response to comment 7:

Thanks to the reviewers for pointing out this problem; and we have already modified the Figures and the quality of the Figures has been improved as much as possible. (As Figure 1-10)

Comment 8: If cell lines were used in the research, a statement addressing the following points must be included in the Materials and Methods section of the manuscript.

  1. Whether the cell lines have been tested and authenticated
  2. The method by which the cells were tested
  3. How and when the cells were last tested

If cells were obtained directly from a cell bank that performs cell line characterizations and passaged in the user's laboratory for fewer than 6 months after receipt or resuscitation, re-authorization is not required. In these cases, please include the method of characterization used by the cell bank.

Response to comment 8: 

Thanks to the reviewer for pointing out this question. The human brain malignant glioma GBM8401 cells (glioblastoma) was purchased from the Food Industry Research and Development Institute (Hsinchu, Taiwan). Human brain malignant glioma U87MG cells (astrocytoma) were obtained from Bioresource Collection and Research Center (BCRC, Hsinchu, Taiwan). GBM8401 cells were cultured in RPMI medium with supplemental 10% fetal bovine serum (FBS) and 1% penicillin/streptomycin antibiotics. U87MG cells were cultured in MEM medium with supplemental 10% fetal bovine serum (FBS) and 1% penicillin/streptomycin antibiotics. Both cell lines were incubated at 37 °C with 5% CO2. Resuspend the cell pellet in a fresh growth medium and add fresh medium to new culture vessels 2 to 3 times a week.

(As M&M; Page line 430 -Page line 437)

Additional clarifications

In addition to the above comments, all spelling and grammatical errors pointed out by the reviewers have been corrected.

We look forward to hearing from you in due time regarding our submission and to responding to any further questions and comments you may have.

Sincerely,

Prof. Tai-Hsin Tsai

Department of Neurosurgery, Kaohsiung Medical University Hospital

No. 100, Tzyou 1st Road, Sham-min District, Kaohsiung City, Taiwan

Reviewer 3 Report

Manuscript Number: IJMS-2158132

Full title: RTA-dh404 Induces Cell Cycle Arrest, Apoptosis and Autophagy in Glioblastoma Cells

Article type: full paper.

Authors: Tai-Hsin Tsai, Yu-Feng Su, Cheng-Yu Tsai, Chieh-Hsin Wu, Kuan-Ting Lee, Yi-Chiang Hsu*

In this article, the authors describe the therapeutic properties of the oleanolic acid derivative RTA-dh404 against glioblastoma cells. After reading the article and reviewing both the existing literature and the literature cited by the authors, one must certainly conclude that this is research on a very promising candidate. Likewise, the procedure and methodology followed in the research reflected in the manuscript are adequate and allow us to clearly reach the conclusions they show (induction of cell cycle arrest, apopotosis and autophagy of glioblastoma cells). Therefore, taking into account that the subject matter, methodology and bibliography are more than correct and deserve publication in the International Journal of Molecular Sciences, I have some observations on some typographical errors that, once removed from the manuscript, will surely result in an easy-to-read article. Some of the ones I have noticed are listed below:

1.- The formatting of the molecular structures in Figure 1 is really flawed. There are different formats, bond lengths and sizes for the same skeleton such as the oleanane syntetic triterpenoids represented. In addition, the authors can eliminate many of the structures depicted, as the number of structures cited throughout the introduction is far less than those depicted. This would allow the structures of RTA-dh404 to be made larger. Furthermore, it should be noted that throughout the text it is referred to as "RTA-dh404", whereas in Figure 1 itself it is referred to as "RTA dh404". The same is true for many structures. I would also ask the authors to take into account and correct that the nomenclature of the first three triterpenoids is confusing, since they indicate a double substitution for the same R substituent. In any case, this figure should be extensively modified.

2.- In line 4 of the first paragraph of section 2.1, the authors forgot to indicate that PrestoBlue(TM) is a registered trademark. In this same section, the authors could include that the viability of GBM8401 and U87MG cells after treatment with RTA-dh404 does not only decrease in a "dose-dependent manner", but also in a "time-dependent manner".

In Figure 2, the authors indicate U87MG cells as "U87-MG". I would ask them to unify the nomenclature if possible. The same applies to figures 3, 4, 5, 6 and 7.

4.- Although "PI" is a widely used abbreviation for propidium iodide, the authors should define it beforehand before using it in the second line of section 2.2.

5.- In Figures 4 and 5, the authors should indicate that the numbers "2", "4" and "8" correspond to the micromolar concentration of RTA-dh404, as they do, for example, in Figure 6.

6.- In the first paragraph on page 7 of the manuscript, the authors forgot the full stop, although the same is true for the titles of several sections. Some have full stops and some do not. Please, I would ask the authors to unify the format.

7.- Probably due to a typo, the authors have the identification of some antibodies such as PARP and cleaved-PARP.

8.- I would ask the authors, if possible, to increase the resolution or improve the legibility of the font in Figure 7. At least in my PDF version of the manuscript, I found it difficult to read.

9.- In the discussion section, in line 3, the authors define RTA-dh404 as a semi-synthetic molecule, when in the abstract they indicate that it is synthetic and again contrast their definition in the first line of the second paragraph on page 2, where they say that it is synthetic. Throughout the manuscript, it is not really clear what the nature of this triterpene is: whether it is a totally synthetic compound or whether it is a modified natural product.

10.- Again, in line 284 of the manuscript, the authors do not emphasise the time dependence of the effect of RTA-dh404 that they have experimentally verified.

11.- In line 285 of the manuscript, the authors should indicate "Annexin-FITC/PI assay..." instead of "annexin-FITC/PI assay...".

12.- On line 376, please indicate correctly the stoichiometric coefficient of oxygen in carbon dioxide as a subscript.

13.- Line 429: "Beckman Coulter..." instead of "Beckman coulter", as is correctly indicated on line 414.

14.- Line 433: please check the typography of RTA-dh404.

15.- Line 440: "... samples were protected..." instead of "... samples protected were...".

16.- Line 477: correct the space between "mean" and the plus-minus sign.

17.- In reference 6, please indicate the stoichiometric coefficient of chlorine in carbon tetrachloride as a subscript.

I have probably overlooked some typos, so I would be grateful if, if possible, the authors could make an extensive revision of the manuscript for publication.

In any case, as can be seen, these are cosmetic remarks to a scientifically well done paper, so I can only recommend its publication after the minor modifications suggested above.

Author Response

Department of Neurosurgery

Kaohsiung Medical University Hospital

Kaohsiung, Taiwan

January 19th, 2023

Dear Reviewer 3:

Thank you for giving me the opportunity to submit a revised draft of our manuscript which is titled: RTA dh404 Induces Cell Cycle Arrest, Apoptosis, and Autophagy in Glioblastoma Cells. We appreciate the time and effort that you and the reviewers have dedicated to providing your valuable feedback on our manuscript. We are grateful to the reviewers for their insightful comments on this paper. We have been able to incorporate changes to reflect most of the suggestions provided by the reviewers. We have highlighted the modification within the manuscript and point-by-point response to the reviewers’ comments and concerns in this revision.

Comments from Reviewer 3:

In this article, the authors describe the therapeutic properties of the oleanolic acid derivative RTA-dh404 against glioblastoma cells. After reading the article and reviewing both the existing literature and the literature cited by the authors, one must certainly conclude that this is research on a very promising candidate. Likewise, the procedure and methodology followed in the research reflected in the manuscript are adequate and allow us to clearly reach the conclusions they show (induction of cell cycle arrest, apoptosis, and autophagy of glioblastoma cells). Therefore, taking into account that the subject matter, methodology, and bibliography are more than correct and deserve publication in the International Journal of Molecular Sciences, I have some observations on some typographical errors that, once removed from the manuscript, will surely result in an easy-to-read article. Some of the ones I have noticed are listed below:

Comment 1: The formatting of the molecular structures in Figure 1 is really flawed. There are different formats, bond lengths, and sizes for the same skeleton such as the oleanane synthetic triterpenoids represented. In addition, the authors can eliminate many of the structures depicted, as the number of structures cited throughout the introduction is far less than those depicted. This would allow the structures of RTA-dh404 to be made larger. Furthermore, it should be noted that throughout the text it is referred to as "RTA-dh404", whereas in Figure 1 itself, it is referred to as "RTA dh404". The same is true for many structures. I would also ask the authors to take into account and correct that the nomenclature of the first three triterpenoids is confusing since they indicate a double substitution for the same R substituent. In any case, this figure should be extensively modified.

Response to comment 1:

Thank you for pointing this out. We agree with this comment. We uniformly changed to RTA dh404. Several novel triterpenoids have been synthesized from oleanolic acid. Further derived CDDO-imidazolide (CDDO-Im), CDDO-ethyl amide (CDDO-EA), CDDO-trifluoroethyl amide (CDDO-TFEA), and CDDO-dihydro-trifluoroethyl amide (CDDO-dhTFEA or RTA dh404). We use RTA dh404 which is a compound extended from CDDO-TFEA (RTA 404).

 (As Figure 1; Page line 54 -Page line 57)

Comment 2: In line 4 of the first paragraph of section 2.1, the authors forgot to indicate that PrestoBlue(TM) is a registered trademark. In this same section, the authors could include that the viability of GBM8401 and U87MG cells after treatment with RTA-dh404 does not only decrease in a "dose-dependent manner", but also in a "time-dependent manner"

Response to comment 2:

Thank you for this suggestion. We have added PrestoBlue(TM) and modified treatment with RTA-dh404 does not only decrease in a dose-dependent manner, but also in a time-dependent manner.

(As Figure 2; Page line 89-Page line 95)

Comment 3: In Figure 2, the authors indicate U87MG cells as "U87-MG". I would ask them to unify the nomenclature if possible. The same applies to figures 3, 4, 5, 6, and 7.

Response to comment 3:

Thank you for pointing this out. We agree with this comment. We have corrected and unified it as U87MG. (As Figure 3-7)

Comment 4: Although "PI" is a widely used abbreviation for propidium iodide, the authors should define it beforehand before using it in the second line of section 2.2.

Response to comment 4:

Thank you for this suggestion. We agree with this comment. PI is based on the principle that apoptotic cells, among other typical features, are characterized by DNA fragmentation and, consequently, loss of nuclear DNA content. (As Figure 3; Page line105 -Page line115)

Comment 5: In Figures 4 and 5, the authors should indicate that the numbers "2", "4" and "8" correspond to the micromolar concentration of RTA-dh404, as they do, for example, in Figure 6.

Response to comment 5:

Thank you for this suggestion. We agree with this comment. In Figure 4, we observed that high-dose of RTA dh404 induced caspase-3 activation. In Figure 5, we observed that RTA dh404 increased the number of cells in the G2/M phase with concentration. (As Figure 4 and Figure 5)

Comment 6: In the first paragraph on page 7 of the manuscript, the authors forgot the full stop, although the same is true for the titles of several sections. Some have full stops and some do not. Please, I would ask the authors to unify the format.

Response to comment 6:

Thanks to the reviewers for pointing out this problem. We have corrected the full-stop problem and rearranged the wording. (As Result2.5.; Page line 166 -Page line 183)

Comment 7: Probably due to a typo, the authors have the identification of some antibodies such as PARP and cleaved-PARP.

Response to comment 7:

Thanks to the reviewers for pointing out this problem. We modified the PARP antibody labeling. (As Result 2.5.; Page line 169 -Page line 179)

Comment 8: I would ask the authors, if possible, to increase the resolution or improve the legibility of the font in Figure 7. At least in my PDF version of the manuscript, I found it difficult to read.

Response to comment 8:

Thanks to the reviewers for pointing out this problem; and we have already modified the Figures and the quality of the Figures has been improved as much as possible. (As Figure 7)

Comment 9: In the discussion section, in line 3, the authors define RTA-dh404 as a semi-synthetic molecule, when in the abstract they indicate that it is synthetic and again contrast their definition in the first line of the second paragraph on page 2, where they say that it is synthetic. Throughout the manuscript, it is not really clear what the nature of this triterpene is: whether it is a totally synthetic compound or whether it is a modified natural product.

Response to comment 9:

Thank you for this suggestion. We agree with this comment. We performed a modified RTA dh404, a synthetic oleanolic acid compound. Furthermore, RTA dh404 and its analogs are oleanolic acid-derived synthetic triterpenoid compounds. (As Discussion; Page line314 -Page line315)

Comment 10: Again, in line 284 of the manuscript, the authors do not emphasize the time dependence of the effect of RTA-dh404 that they have experimentally verified.

Response to comment 10:

Thank you for pointing this out. We agree with this comment. Our experimental results observed that RTA dh404 inhibited cell growth in a dose and time-dependent manner. (As Figure 2; Page line 86-Page line 95)

Comment 11: In line 285 of the manuscript, the authors should indicate "Annexin-FITC/PI assay..." instead of "annexin-FITC/PI assay...".

Response to comment 11:

Thank you for this suggestion. We agree with this comment. We modified it to Annexin-FITC/PI assay. (As Discussion; Page line 334 -Page line 334)

Comment 12: On line 376, please indicate correctly the stoichiometric coefficient of oxygen in carbon dioxide as a subscript.

Response to comment 12:

Thank you for this suggestion. We corrected the stoichiometry of oxygen in carbon dioxide. (As M&M Cell culture; Page line 440 -Page line 441)

Comment 13: Line 429: "Beckman Coulter..." instead of "Beckman coulter", as is correctly indicated on line 414.

Response to comment 13:

Thank you for this suggestion. We have modified it to FACSCalibur, BD Pharmingen. (As M&M Caspase-3 activity assay; Page line 488 -Page line 489)

Comment 14: Line 433: please check the typography of RTA-dh404.

Response to comment 7:

Thanks to the reviewers for pointing out this problem, and we uniformly changed it to RTA dh404. (As M&M; Page line 526 -Page line 539)

Comment 15: Line 440: "... samples were protected..." instead of "... samples protected were...".

Response to comment 15:

Thanks to the reviewers for pointing out this problem. We didn't use this experiment in our manuscript, so we decided to remove this method. (As M&M; Page line 526 -Page line 539)

Comment 16: Line 477: correct the space between "mean" and the plus-minus sign.

Response to comment 16:

Thanks to the reviewers for pointing out this problem. We corrected the space between "mean" and the plus-minus sign.

(As M&M Data analysis; Page line527 -Page line527)

Comment 17: In reference 6, please indicate the stoichiometric coefficient of chlorine in carbon tetrachloride as a subscript.

Response to comment 17:

Thanks to the reviewers for pointing out this problem. We corrected the stoichiometric coefficient of chlorine in carbon tetrachloride as a subscript (As Reference 6; Page line 572-Page line 574)

Additional clarifications

In addition to the above comments, all spelling and grammatical errors pointed out by the reviewers have been corrected.

We look forward to hearing from you in due time regarding our submission and to responding to any further questions and comments you may have.

Sincerely,

Prof. Tai-Hsin Tsai

Department of Neurosurgery, Kaohsiung Medical University Hospital

No. 100, Tzyou 1st Road, Sham-min District, Kaohsiung City, Taiwan

Round 2

Reviewer 1 Report

Revision accepted, thank you

Author Response

Thank you very much.

Reviewer 2 Report

Accept after minor revision (corrections to minor methodological errors and text editing)

Author Response

Thank you very much.